# Pancreatitis in Pregnancy—Comprehensive Review

**DOI:** 10.3390/ijerph192316179

**Published:** 2022-12-03

**Authors:** Agnieszka Mądro

**Affiliations:** Department of Gastroenterology with Endoscopic Unit, Medical University, 20-059 Lublin, Poland; agnieszka.madro@wp.pl

**Keywords:** acute pancreatitis, chronic pancreatitis, cholelithiasis, hypertriglyceridemia, pregnancy

## Abstract

Acute and chronic pancreatitis, until recently observed incidentally in pregnancy, has occurred much more frequently in the last 2–3 decades. Particularly severe complications for the mother and fetus may be a consequence of acute pancreatitis. Therefore, it is important to know more about the diagnostic and therapeutic possibilities of pancreatic diseases in the course of pregnancy. Epidemiology, causes, clinical characteristics, differential diagnosis, and complex management are presented in this review. Particular emphasis is on the prevention of acute pancreatitis (AP) through the proper diagnosis and treatment of cholelithiasis and hypertriglyceridemia, both before and during pregnancy. The most up-to-date reports and management strategies are presented. This publication contributes to a wide group of scientists and practitioners better understanding the discussed issues, and indicates the directions of research for the future.

## 1. Introduction

Pregnancy is a special time in a woman’s life, when waiting for a new life is accompanied by fear for its proper development. A noticeable, but also confirmed by researchers, trend is the increasingly late age of women when they decide to procreate. This has ramifications for both the mother and baby. Pancreatic diseases, observed incidentally during pregnancy until recently, have occurred much more frequently in the last 2–3 decades. This is related to many other changes observed in the modern world, including inappropriate eating habits leading to obesity. These changes generate a greater risk of developing various diseases, including pancreatic diseases and especially acute pancreatitis (AP). The management of AP requires the consideration of physiological and anatomical changes during pregnancy in conjunction with the local and systemic effects of AP. More problems arise when endoscopic or surgical intervention is urgently needed, as both have a potentially serious risk to the mother and the fetus.

Therefore, it is important to know more about the diagnostic and therapeutic possibilities of pancreatic diseases in the course of pregnancy.

This comprehensive review was prepared on the basis of the recent literature. Four databases (PubMed, Web of Science, MEDLINE, and Embase) were searched. The following search keywords were used separately and in combination: acute pancreatitis, chronic pancreatitis, cholelithiasis, hypertriglyceridemia, and pregnancy.

## 2. Epidemiology of Acute Pancreatitis in Pregnancy

Acute pancreatitis in pregnant women occurs with a frequency of 1/1000 to 1/5000 pregnancies [1,2]. Due to significant progress in the prevention, diagnosis, and proper treatment, the death rate of pregnant women due to AP, once very high (37%), has dropped to 3.3%, and fetal mortality from almost 60% to 11.6–18.7%, according to various reports [1,3]. The mean age of onset of AP in pregnancy was 28.5 years. AP was most common in the third trimester of pregnancy. About one-third of the women with AP develop severe pancreatitis. Mortality in pregnant women with acute pancreatitis in pregnancy is comparable to the rate in the general population, but the cumulative maternal death rate was the highest in the first trimester at 12.7%, compared with 7.9 and 6.4% in the second and third trimesters, respectively. The same data were obtained for fetal deaths: the highest death rate was recorded in the first trimester (20.9%). Intrauterine fetal death was the most common in the third trimester (8.8%), while stillbirths were highest in the second trimester (6.2%) [3].

## 3. Etiological Factors of AP in Pregnancy

There are many risk factors for the development of acute pancreatitis in pregnancy that should be identified before or during the first weeks of pregnancy. These include gallstone disease; hyperlipidemia, especially hypertriglyceridemia; subsequent pregnancies; obesity; a high-fat diet [4].

In contrast to the general population, where gallstone disease and alcohol abuse are the two most common etiological factors, cholelithiasis comes to the fore in pregnant women. Worryingly, despite the widespread knowledge of alcohol’s detrimental effects on fetal development, alcohol is still a common etiological factor [5]. Hypertriglyceridemia has recently been a common cause of AP [6]. Less frequently observed etiological factors are hyperparathyroidism, infectious agents, medications, or injuries. In some cases, it is not possible to determine the cause of AP [7]. Idiopathic pancreatitis is diagnosed on the basis of clinical and laboratory tests confirming AP after excluding all the above-mentioned causes.

## 4. Gallstone Disease and AP in Pregnancy

Particular attention should be paid to the risk of acute pancreatitis in women diagnosed with cholelithiasis before pregnancy. The possibility for the complications of gallstone disease, including AP, is greater in pregnancy than that in the general population. Pregnancy is also a period that increases the risk of gallstone formation [8]. There is a tendency to change the composition of bile, leading to the bile ducts malfunctioning. The tendency to vomit observed in the first trimester also has an undoubted influence on the formation of bile deposits; in some women, it takes the form of incontinence or vomiting, which promotes dehydration and the thickening of bile [5]. During pregnancy, as a result of the action of hormonal factors, the composition of bile changes, which consists of elevated cholesterol content, the inhibition of the conversion of cholesterol into bile acids, and quantitative changes in the proportion of bile acids. Metabolic disorders such as insulin resistance and elevated leptin levels are also involved in the formation of gallstones [9]. As a result of the action of progesterone, water absorption by the gallbladder mucosa decreases already in the first trimester of pregnancy, which increases its volume and impairs contractility. All these phenomena contribute to the possibility of gallstone formation at the end of the first trimester, and the risk increases significantly in the second and third trimesters of pregnancy [10].

Therefore, to assess the gallbladder and bile ducts, it is recommended to perform an ultrasound of the abdominal cavity at the end of the first trimester of pregnancy; the next ones should be performed at the end of the second trimester of pregnancy [5].

## 5. Alcohol and AP during Pregnancy

Although alcohol’s detrimental effects on fetal development are well-known, there are reports of acute pancreatitis of this etiology. In caring for a pregnant woman, special attention is paid to emphasizing the need to give up drinking alcohol during pregnancy and breastfeeding. Unfortunately, many people still underestimate this limitation, leading to many complications for both the mother and the fetus. In practice, documenting alcohol as a cause is quite difficult; therefore, the diagnosis is most often performed after excluding other causes, family and environmental history, and high levels of gamma-glutamyltranspeptidase (GGTP) in the blood serum [10,11].

## 6. Hypertriglyceridemia and AP during Pregnancy

Hypertriglyceridemia (serum triglycerides > 150 mg/dL) is a known etiology of acute pancreatitis. During physiological pregnancy, changes in the carbohydrate and lipid metabolism are observed to ensure the greatest possible availability to the fetus: increased glucose production, progesterone synthesis, lipogenesis, and impaired lipolysis. In women with impaired lipoprotein metabolism, these adaptive changes can lead to severe hypertriglyceridemia. In physiological pregnancy, triglyceride levels increase by 2–4 times in the third trimester, but rarely exceed 300 mg/dL. The increased risk of AP is above 500 mg/dL, but it is most common at triglyceride levels above 1000 mg/dL. The risk of AP increases with the progression of pregnancy: 19% in the first trimester, 26% in the second trimester, and 53% in the third [10]. It is recommended to measure the lipid profile of women in early pregnancy if a family member has a history of hyperlipidemia. If a patient is diagnosed with high lipid values, AP prophylaxis is initiated by following a diet with a fat content of less than 20% and enriched with omega-3 acids, especially docosahexaenoic acid (DHA). The use of insulin enhances the action of lipoprotein lipase and leads to the degradation of chylomicrons, which in turn reduces the level of triglycerides. On the other hand, the use of heparin stimulates the release of lipoprotein lipase. Statins must not be used during pregnancy because they have a proven teratogenic effect on the fetus, and other drugs such as fibrates, cholestyramine, and niacin should not be administered to pregnant women due to a lack of safety studies [12]. There are reports in the literature about the beneficial effect of the use of plasmapheresis in the case of high triglycerides in the prevention of AP in pregnant women [6]. Zeng et al. suggested the use of mean platelet volume (MPV) levels as a predictor of severe AP in pregnant women with hypertriglyceridemia, but such results require additional large-scale prospective research [13].

In women diagnosed with hypertriglyceridemia before pregnancy, it is recommended to change the lifestyle (low-fat diet, omega-3 acids, physical activity, alcohol abstinence), to control other adverse factors (diabetes), and to avoid drugs that may cause acute pancreatitis (glucocorticoids, estrogens). Significant weight gain [14] should be avoided during pregnancy. Gupta et al. suggested that high-risk women check their triglyceride levels once each trimester. Fasting triglycerides > 250 mg/dL should prompt monthly triglyceride levels, screening for gestational diabetes, implementing a strict low-carbohydrate, low-fat diet, and exercising. Fasting triglycerides > 500 mg/dL, despite stringent dietary and lifestyle modifications, should prompt treatment with omega-3 fatty acids and continuing a low-fat diet (<20 g fat/day or <15% calories) under the guidance of a registered dietitian. Plasmapheresis should be considered early in asymptomatic pregnant women with fasting triglycerides > 1000 mg/dL or in pregnant women with clinical symptoms of pancreatitis and triglycerides > 500 mg/dL despite maximal lifestyle changes and pharmacological treatment [15,16].

## 7. Clinical Symptoms and Diagnosis of AP

The diagnosis of acute pancreatitis is based on the fulfillment of two out of the three following criteria: clinical picture with epigastric pain radiating to the back, pancreatic amylase activity > 3 times the upper limit of normal (blood amylase activity may normalize within 48–72 h from the onset of symptoms; for a long time, the activity of amylase in the urine and the isoenzyme of pancreatic amylase in the blood are maintained at a higher level), and features of acute pancreatitis in imaging tests [17]. The diagnosis of AP should be confirmed within the first 48 h from admission to the hospital [17].

The triad of characteristic symptoms of AP (severe girdling localized mainly in the epigastric region, abdominal pain radiating to the back; nausea and vomiting that does not bring relief; paralytic intestinal obstruction) leads to the suspicion of acute pancreatitis in pregnant women. These symptoms may be associated with low-grade fever. In the case of biliary etiology, the first symptoms may be colic pains in the right hypochondrium, vomiting, poor tolerance for fatty foods, and even jaundice [17].

Physical examination in pregnant women is difficult, especially in advanced pregnancy, and should be performed with particular care. In the case of acute pancreatitis, attention should be paid to the pressure pain of the abdominal cavity of a diffuse nature, lowered or no peristalsis, and the yellowing of the skin layers. The diagnosis of acute pancreatitis is based on the determination of serum and urine amylase activity, and serum lipase activity. Other routinely performed laboratory tests contain morphology, electrolytes, urea, creatinine, glucose, C-reactive protein (CRP), bilirubin, alanine aminotransferase (ALT), aspartate aminotransferase (AST), alkaline phosphatase, gamma-glutamyltranspeptidase (GGTP). In the first 24–48 h, prognostic factors that may indicate the possibility of a severe course of the disease should be assessed [1].

Imaging diagnostics is based primarily on the ultrasound of the abdominal cavity (AUS); its main purpose is to confirm or rule out biliary etiology and possible complications of acute pancreatitis such as pancreatic parenchyma necrosis, inflammatory infiltrates, or fluid reservoirs. The test is safe for the mother and the fetus, but has limited usefulness in the case of late pregnancy, obesity, or the presence of significant amounts of intestinal gas. A recent Cochran database reported that AUS has an average sensitivity of 73% and a specificity of 91% for the detection of bile duct stones [18]. If gallstones are suspected, endoscopic ultrasonography (EUS) is very helpful and can be performed at all reference centers. EUS is particularly sensitive in detecting microlithiasis and sludge in bile. However, its role in pregnancy has not been thoroughly studied in the literature. Small published retrospective studies indicated that EUS is effective and safe in pregnancy at any gestational age. EUS does not involve radiation and can save pregnant patients from having an unnecessary invasive procedure such as endoscopic retrograde cholangiopancreatography (ERCP) [8]. Magnetic resonance imaging (MRI) is characterized by high sensitivity and specificity in the diagnosis of choledocholithiasis with a reported accuracy of more than 90% [19]. These facts render MRI an attractive method of imaging in pregnancy. However, recently, some reports have raised concerns about the effects of heat on the fetus. MRI performance should be limited to the first trimester of pregnancy, as it may cause tissue overheating and consequently impaired fetal development [20]. Abdominal computed tomography (CT) has 72–78% sensitivity and 96% specificity in the diagnosis of bile duct stones [21]. However, its role in pregnancy is severely limited due to the potential effect of ionizing radiation and contrast on the fetus; therefore, it is recommended to avoid abdominal computed tomography throughout pregnancy unless the benefits of the study outweigh the risks to the fetus (teratogenic effect) [8,10].

The most important task after the diagnosis of AP is to predict its severity. AP was divided into three categories according to severity on the basis of the Revised Atlanta criteria: Mild acute pancreatitis (MAP) referring to pancreatitis without organ dysfunction or generalized complications. Moderately severe pancreatitis (MSAP) refers to pancreatitis with persistent organ dysfunction or localized/generalized complications within 48 h of starting treatment. Severe pancreatitis (SAP) refers to pancreatitis with persistent organ dysfunction or localized/generalized complications for more than 48 h after treatment [22]. Organ dysfunction is defined on the basis of the modified Marshall scoring [22]. Treatment modality depends on the predicted severity of AP.

The same scales as those in the general population (Ranson, Bisap, Apache II) should be used to predict the course of AP. The choice of scale usually depends on the experience of the center [17,23,24]. In recent years, researchers have highlighted the usefulness of lactate dehydrogenase (LDH) determination, especially the LDH-4 isoenzyme, in the early prognosis of AP severity. A similar value is assigned to the red target width of the distribution (RDW); values above 14.35 are associated with a risk of severe course. It is also worth paying attention to other, single prognostic factors facilitating the diagnosis of a potentially severe course of AP, such as CRP, hematocrit, interleukin 6 (determined on days 1 and 2 of the disease), and BMI (a high level suggests the risk of a severe course of AP) [1,25]. Recently, Yang et al. defined a prediction model for moderately severe and severe AP in pregnancy [26]. Researchers identified four developed predictors and established a predictive nomogram model. This nomogram includes lactate dehydrogenase, triglyceride, cholesterol, and albumin levels as independent predictors. These tests are routinely used in clinical practice. The proposed nomogram seems simple, but has some limitations: too-small patient groups, one research center, retrospective study, and the good predictive ability of moderately severe and severe AP in pregnancy, but no further differentiation between moderate and severe AP can be achieved. The prognosis of moderate AP is not as bad as that of severe AP [26]. Sheng et al. created a nomogram for the prediction of persistent organ failure (POF) in AP in pregnancy [27]. One-way and multivariate logistic regression analysis showed that lactate dehydrogenase, triglycerides, serum creatinine, and procalcitonin were independent risk factors in predicting POF in APIP. The POF nomogram was created using four indicators. The area under the curve was 0.875 (95% CI: 0.80–0.95) [27].

## 8. Differential Diagnosis

Other conditions with severe abdominal pain, nausea, and vomiting should be considered in the differential diagnosis: myocardial infarction, gastric ulcer and duodenal ulcer, acute appendicitis, acute cholecystitis and/or biliary tract inflammation, acute mesenteric ischemia, urinary tract inflammation gastrointestinal and obstetric complications: acute fatty liver, metastatic disease, HELLP, placental detachment or uterine rupture [10].

## 9. Comprehensive Management of AP in Pregnant Women

Due to the special situation, which is the need to care for both the mother and the fetus, a multidisciplinary team is needed consisting of an obstetrician, gastroenterologist, anesthesiologist, and surgeon. It is necessary to monitor the health of the baby and mother during pregnancy, childbirth, and puerperium. Management should depend on both the severity of acute pancreatitis and the stage of pregnancy [28]. Conservative management in the first trimester is recommended, and in the case of acute biliary pancreatitis. This consists in adequate hydration, the administration of antispasmodics (drotaverine, hyoscine butylbromide) and analgesics (paracetamol), and the correction of electrolyte disturbances. Painkillers that cannot be used during pregnancy include nonsteroidal anti-inflammatory drugs in the first and third trimesters of pregnancy, acetylsalicylic acid (throughout pregnancy), metamizole, and opioids [29].

It is recommended to start oral nutrition on a low-fat diet as soon as possible (after 1–2 days of fasting without limiting the consumption of neutral fluids, and up to 3 days from the onset of symptoms) to start enteral nutrition in the event of a prognosis of severe acute pancreatitis. Antibiotics are reserved for cases of infected pancreatic necrosis, cholangitis, or other inflammatory conditions (urinary tract, lungs) [17].

Surgical intervention during pregnancy has some limitations. The first limitation is the potentially adverse events of general anesthesia (GA). The exact short- and long-term effects of GA on the fetus are still unknown, and general anesthetics appear to have the ability to cross the placenta [30]. Any type of GA surgical intervention in pregnant women is difficult due to the accompanying anatomical and physiological changes. Apart from deciding whether to perform a cholecystectomy, a big problem is deciding how to perform it: open or laparoscopic. Sedaghat et al. described the potential limitations associated with the use of laparoscopy in pregnancy to technical and nontechnical problems, especially in the third trimester: the risk of the manipulation of the uterus by the umbilical part, poor vision and limited laparoscopic access due to the pregnant uterus, the physiological impact of pneumoperitoneum on the fetus, and hypercapnia [31]. Compared to open surgery, laparoscopic intervention in pregnancy is associated with a significant reduction in the time of surgery, the length of stay in a hospital, wounds and other complications, postoperative pain, the incidence of thromboembolic complications, postoperative intestinal obstruction, and a faster return to normal functioning [32]. On the basis of literature guidelines, it is recommended that the second trimester of pregnancy is the best time to perform laparoscopic cholecystectomy when necessary [23,24].

The performance of endoscopic retrograde cholangiopancreatography (ERCP) in pregnancy is controversial. The indication for ERCP in pregnancy is symptomatic cholelithiasis manifested by jaundice, cholangitis, or biliary pancreatitis. Recent data, although limited by the lack of information on trimester, suggest that early intervention (ERCP or cholecystectomy) is preferable to no treatment [33,34]. Still, many researchers support the notion that, in most cases, the introduction of one or more plastic bile stents for biliary decompression during ERCP without fluoroscopy prevents biliary obstruction. Definitive therapy should be performed after childbirth [35]. ERCP with fluoroscopy carries a risk of teratogenic radiation to the fetus and an increased risk of future malignancies in the baby. It is safest to perform ERCP without radiation exposure during the third trimester of pregnancy. Due to the high risk for the fetus, ERCP in pregnancy should be considered in a reference center with a team with extensive experience and in performing tests without the use of fluoroscopy, possibly with choledochoscopy. The preterm delivery of pregnancy should be considered in the case of a worsening clinical condition of the mother despite 24–48 h intensive treatment, paralytic ileus, imminent miscarriage, intrauterine fetal death, fetal defects, and severe course of AP [36]. 

A schematic algorithm concerning the management of AP in pregnant women is presented in Figure 1.

### 9.1. Effect of AP on the Fetus

Acute pancreatitis in pregnancy carries an increased risk of preterm labor, prematurity, and death. The effects of the disease on the fetus are related, on the one hand, to its course and, on the other hand, to the performed diagnosis or treatment. The consequence of the severe course of acute pancreatitis accompanied by SIRS may be the disruption of the functioning of the placenta, which may result in the death of the child [3,37]. On the other hand, diagnostics with the use of radiation, anesthesia, and pharmacotherapy may have a potentially harmful effect on the fetus. However, we still do not have large, reliable studies that would allow for a precise estimation of this risk. Hence, it is so important to care for a pregnant woman in such a way that (thanks to early diagnosis), it is possible to prevent the development of acute pancreatitis, and if the disease develops, it could be properly treated. Children born to mothers diagnosed with hypertriglyceridemia during pregnancy have impaired lipid homeostasis and are more prone to atherosclerosis [38].

Recently, Praven Kumar et al. conducted a large systematic review of 16 studies involving 8466 pregnant patients [36]. The overall prevalence of AP ranged from 0.225/1000 to 2.237/1000 pregnancies. Gallstone disease was the most common cause, ranging from 14.29 to 96.3%, with eastern studies reporting more cases of hypertriglyceridemia as the etiology. Mild pancreatitis has been reported in 33.33–100% of cases with a milder disease in Western studies. The incidence of AP was higher in the third trimester (27.27–95.24%). Maternal mortality ranged from 0 to 12.12/100 pregnancies. Fetal loss ranged from 0 to 23.08%, and adverse effects on the fetus ranged from 0 to 57.41%. Neonatal mortality ranged from 0 to 75.5/1000 live births of newborns [36].

In conclusion, acute pancreatitis in pregnant women is a rare but potentially serious complication, and even a life-threatening disease for the mother and fetus. The greatest risk of complications is associated with acute pancreatitis caused by hypertriglyceridemia [15]. The pregnancy should be terminated by experienced obstetricians and gastroenterologists depending on the condition of the fetus and mother.

### 9.2. Chronic Pancreatitis (CP) 

Chronic pancreatitis in pregnant women is rarely observed; in addition, some cases of CP are classified as acute pancreatitis (due to an increase in the activity of pancreatic enzymes in exacerbation of the disease), but no reliable statistical studies are available in the literature. Some of the publications date back to the early 21st century, when genetic testing was not conducted. This is still the case outside reference centers. Hence, the main cause of CP in pregnant women is alcohol abuse, and the second is idiopathic CP, but it should be assumed that, in both of these groups, mutations predisposing to CP can be found in the majority of women [38].

On the basis of the limited literature, the frequency of CP exacerbations in pregnancy is lower than that in the period before and after pregnancy. Women diagnosed with CP at a young age became pregnant later and had fewer children compared to those diagnosed with CP at a later age. There were no significant differences in the course of pregnancy in both groups with regard to gestational diabetes, arterial hypertension, or premature delivery. There was also no evidence of an increased number of miscarriages, low birth weight, or birth defects [39,40].

Women with CP who become pregnant should be monitored by both an obstetrician and gastroenterologist, and consulted by a dietitian and pain management specialist. If CP is diagnosed before pregnancy, diagnostic tests in this direction are not required, bearing in mind that the most sensitive imaging tests (CT) are contraindicated in pregnancy. In doubtful cases, it is useful to perform EUS and determine fecal elastase-1. Due to the possibility of complications in the form of cysts, it is recommended to regularly (every 2–3 months) perform an ultrasound of the abdominal cavity. The most important element of the treatment is the appropriate supplementation of pancreatic enzymes in the form of enteric capsules containing microspheres sensitive to the surrounding pH, and micropellets with a high concentration of lipase. The recommended dose for main meals is 30,000–40,000 lipase units and half the dose for snacks. Limiting fat intake is not recommended unless severe diarrhea persists despite adequate enzyme substitution. Vitamin D3 supplementation is also recommended, especially in severe exocrine pancreatic insufficiency. Severe pain is a predominant problem in pregnant women with CP and often the first symptom of the disease. For the treatment of pain in CP (the most commonly observed in exacerbations), the same drugs are recommended as for the treatment of acute pancreatitis (the administration of antispasmodics (drotaverine, hyoscine butylbromide) and painkillers (paracetamol)). Diabetes management in the course of CP in pregnant women requires special attention, as it may be unstable, requiring rapid therapeutic interventions [41].

A special form of CP is autoimmune pancreatitis (ACP); in doubtful cases (especially in the presence of jaundice), diagnostics should be extended to include the determination of IgG4, antinuclear antibodies (ANAs), and against carbonic anhydrase. The diagnosis of ACP fundamentally changes the management, and the rapid initiation of steroid therapy, which is not contraindicated in pregnancy, may contribute to rapid clinical improvement [40]. Complications that may occur in the course of CP require a careful, individual approach, taking into account the guidelines recommended in such cases in CP without pregnancy. In a situation where a pregnant woman has abdominal pain for no apparent reason, diagnostics for the insufficiency of the exocrine pancreas (determination of elastase 1 in the stool) should always be performed and, if confirmed, enzymatic supplementation should be used. In conclusion, CP is usually diagnosed before pregnancy and requires the careful guidance of the pregnant woman by a multidisciplinary team to minimize potential adverse effects on the mother and baby.

## 10. Conclusions

Pancreatitis in pregnant women, especially acute pancreatitis, is a challenge for both obstetricians and gastroenterologists. Acute pancreatitis can be potentially life-threatening to both the mother and the fetus. Therefore, risk factors such as gallstones and hypertriglyceridemia should be diagnosed early in pregnancy, which can be achieved with simple tests. Each pregnant woman with abdominal pain should have pancreatic enzymes tested to confirm or exclude AP. When performing diagnostic and therapeutic procedures, their side effects on the fetus should be taken into account, and particularly risky ones should be undertaken only when the potential benefits outweigh the risks. For this reason, a pregnant woman with pancreatitis should be treated in a reference center with all diagnostic possibilities and an experienced team.

This publication contributes to a better understanding of the discussed issues by a wide group of scientists and practitioners, and indicates the direction of research for the future.

## Figures and Tables

**Figure 1 ijerph-19-16179-f001:**
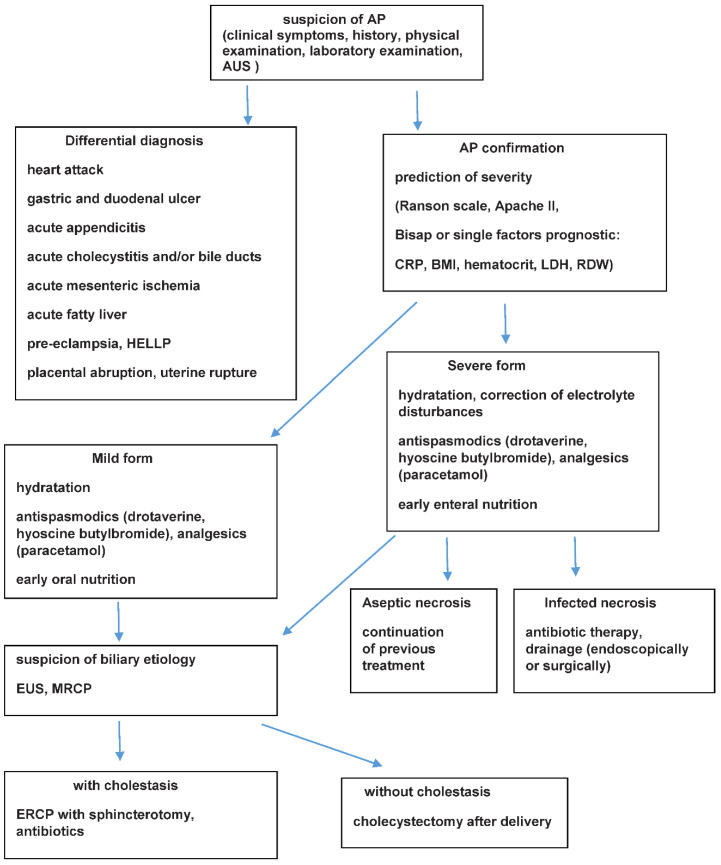
Schematic algorithm concerning management of AP in pregnant women. AP (acute pancreatitis); AUS (abdominal ultrasonography); BMI (body mass index); CRP (C-reactive protein); ERCP (endoscopic retrograde cholangiopancreatography); EUS (endoscopic ultrasound); HELLP (hemolysis, elevated liver enzymes low platelts); LDH (lactate dehydrogenase); MRCP (magnetic resonance cholangiopancreatography); RDW (red cell distribution width).

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
