# Peer review of "Pancreatitis in Pregnancy—Comprehensive Review"

_ijerph, 2022, doi:10.3390/ijerph192316179_

Round 1

Reviewer 1 Report

I thank the author for this article on pancreatitis in pregnant women. The paper is well written, but it does not present any novelty compared to other articles already in the scientific literature, and most importantly, there is a sparse bibliography.

Author Response

Thank you for your review and opinion that the paper is well written. So far, I have not found such a comprehensive study in the literature, which is why this article was created. It evaluates different aspects of pancreatitis in pregnancy from a double view of both the mother and the fetus. The manuscript is addressed especially to obstetricians for whom pancreatic diseases in pregnant women can be a challenge.

Reviewer 2 Report

It is advisable for a better reading of the work to add some tables or figures.

For example, it is possible to summarize in a table, the clinical and / or laboratory data of the various patients present in the various paper.

This work is very interesting because it evaluates different aspects of pancreatitis in pregnancy. It has a double view on both the mother and the fetus. It also deals with both pharmacological and surgical aspects. It also sums up all the laboratory tests quite well, both the well-known ones and the more research ones. In the literature there are several works that deal with these aspects, but this seems to me an excellent summary of the different reviews. The only criticality already highlighted is the lack of tables or figures for easy reading. Grouping the various data present in the literature can be excellent for allowing a comparison as well. Using a few charts to highlight some of the relevant data in the review can be helpful. While this work can be accepted in this form, I wouldn't mind if the authors made the changes I recommended.

Author Response

Thank you very much for the review, I am happy with the high rating of the article. I agree that including tables or figures would be helpful for easier reading. Therefore, as suggested by the reviewer, a graph presenting the algorithm of management in acute pancreatitis in pregnant women will be included.

Reviewer 3 Report

The “Pancreatic diseases in pregnancy-comprehensive review”” manuscript demonstrates mainly acute pancreatitis in pregnancy. The manuscript has several limitations and the English language is poor, which should be improved.

My suggestion is major revision.

Major comments and questions

1. )The title and the content of manuscript not totally correspond to each other. Based on the title “Pancreatic diseases in pregnancy- comprehensive review” I would expect discussion from endocrine and exocrine disorders of the pancreas in pregnancy. However, the manuscript discusses only AP, and shortly CP and autoimmune pancreatitis, but there is no discussion about pancreatic cancer, gestational diabetes etc. I recommend modifying the title, for example: Pancreatitis in pregnancy – comprehensive review.

2.) As it is a comprehensive review, how was the literature search performed? Where were the literature search done and which were the search terms?

3.) The reading and following the text is difficult. Some parts should be restructured.

4.) “AP was most common in the third trimester of pregnancy. About one-third of the women develop severe pancreatitis”. For the reader this means that one-third of pregnant women in third trimester develop severe AP, but I think is not true (please correct the meaning of the sentences). Furthermore, fortunately AP is not a common disease during pregnancy, it is a rare disease. For example, gestational diabetes is a common disease during pregnancies.

5.) “These include: gallstone disease; hyperlipidaemia, especially hypertriglyceridaemia; woman's age (increases in old age); subsequent pregnancies; obesity; a high-fat diet”. In my understanding, in the reference 4 the woman’s age is a risk factor but only in postpartum period. During pregnancy the reference 4 did not investigated the effect of age on AP development, therefore the age can not be discussed as risk factor during pregnancy.

6.) “Lipid disorders, especially hypertriglyceridemia, have recently been a common cause”. Among lipid disorders only HTG causes AP, and other lipid disorders (eg. hypercholestrinemia) do not cause. Please correct this sentence

7.) During HTG there are novel TG lowering molecular therapies (inhibitors of ANGPLT3 and APOC3: monoclonal antibodies, antisense oligonucleotides, and small interfering RNA silencers). What does the author think about the future use of these therapies during pregnancy?

8.) Clinical symptoms and diagnosis of AP is hard to follow. I recommend to start with diagnostic criteria of AP then discuss the symptoms and diagnostic possibilities in detail.

9.) Severity of AP. You mentioned Atlanta criteria, but this is Revised Atlanta criteria. Furthermore, the “Moderate to severe pancreatitis” is moderately severe pancreatitis based on the Revised Atlanta criteria.

10.) “The pregnancy should be terminated by experienced obstetricians and gastroenterologists depending on the condition of the fetus and mother”. Why should be terminated? I can not see the relationship of this sentence with the previous sentences. Could you please introduce this sentence or make relationship with the previous section, because this is a very serious statement, which involve killing a person (fetus) by abortion.

11.) A summary with some future direction would be useful at the end of the manuscript.

Minor comments and questions:

1. “Pregnancy is also a period that increases the risk of gallstone formation [6,8]”. The REF 6 maybe not the best reference for this sentence

2. The following sentence needs references: “Metabolic disorders such as insulin resistance and elevated leptin levels are also involved in the formation of gallstones”

3. References: a.) There are several sentences which need references. b.) Sometimes the reference is not in the first sentence, but it comes later. Please move references in the first sentences where you mention is. For example:
“Recently, Yang et al. idefined prediction model for moderately sever and sever AP in pregnancy.” I tried to find the reference for this sentence, but it was some sentences later. And the same thing was here: “Sheng et al. created a nomogram for the prediction of persistent organ failure (POF) in AP in pregnancy”

4. “In physiological pregnancy, triglyceride levels increase 2-4 times in the third trimester, but rarely exceed 300 mg / dL” This sentence needs reference

5. “The increased risk of AP is above 500 mg / dL, it also increases with the progression of pregnancy -19% in the first trimester, 26% in the second trimester and 53% in the third, but it is most common at triglyceride levels above 1000 mg% [9]” This sentence is hardly understandable. Please make it clear.

6. This sentence was written two times: “Fasting triglycerides> 250 mg / dL should prompt you to monthly triglyceride levels, screening for gestational diabetes, and a strict low-carbohydrate and low-fat diet and exercise”

7. Subtitle formatting: I think that “Comprehensive management of AP in pregnant women” should be bold as without italic as other titles and this comment refers to other further titles

8. Subtitle “Effect of AP on the Fetus:” The colon is unnecessary at the end of the title

9. “Recently, Praven Kumar et al. conducted a large systematic review of 16 studies involving 8,466 pregnant patients.” This sentence need reference

10. “The greatest risk of complications is associated with acute pancreatitis caused by hypertriglyceridaemia” This sentence need reference

11. “If AP is diagnosed, a pregnant woman should be under the care of a multidisciplinary team consisting of an obstetrician, gastroenterologist, surgeon and anesthesiologist” This sentence is a repetition. You suggested it earlier in the manuscript

Author Response

Thank you very much for your review and all your comments. Most of the comments are justified, and all paragraphs and sentences with reservations will be restructured. My manuscript was undergo extensive English revisions by a native English-speaking colleague.

The reviewer made many major and minor comments, below I am answering all of them.

Major comments and questions:

  1. I agree with the reviewer that the title "Pancreatitis in pregnancy" better reflects the content of the article. Problems related to diabetes are not included in the article, because it is not within my competence - I am a gastroenterologist. Pancreatic cancer, on the other hand, is rarely diagnosed in pregnant women. Over the past 30 years, several cases of pancreatic cancer in pregnant women have been described in the literature, but no guidelines for management have been developed.
  2. This comprehensive review was prepared on the base of recent literature. A search of four databases (PubMed, Web of Science, MEDLINE and Embase) was undertaken. The following search keywords were used, separately and in combination (acute pancreatitis, chronic pancreatitis, cholelithiasis, hypertriglyceridemia, pregnancy).
  3. I will restructure some parts according to the reviewer’s comments.
  4. The sentence has been corrected.
  5. Woman's age as a risk factor for acute pancreatitis was removed.
  6. The sentence has been corrected
  7. The use of novel TG lowering molecular therapies (inhibitors of ANGPLT3 and APOC3: monoclonal antibodies, antisense oligonucleotides, and small interfering RNA silencers) has not yet become widely used in the treatment of hypertriglyceridemia, so I do not think it will be considered for use in pregnancy in the near future. It takes a long time to evaluate both their effectiveness and safety.
  8. The section has been corrected.
  9. The sentence has been corrected.
  10. Sentence “early termination of pregnancy” was taken from the cited literature, but I agree with the Reviewer that it may suggest termination by abortus. After consulting with the obstetrician, I change to preterm delivery.
  11. Summary paragraph was added.

Minor comments and questions:

  1. REF 6 was removed as recommended by the reviewer.
  2. Correct REF was inserted.
  3. REF was inserted
  4. REF was inserted
  5. The sentence has been corrected.
  6. The sentence has been deleted.
  7. The sentence has been corrected.
  8. The colon has been removed.
  9. REF was inserted
  10. REF was inserted
  11. The sentence has been deleted.

Round 2

Reviewer 1 Report

I have already expressed my opinion about the manuscript and even after the changes made I do not find much diversity. I leave the final decision to the editor.

Author Response

Thank you for your review once again and for your opinion. As I mentioned earlier, I have not found such a comprehensive study in the literature addressed to both gastroenterologists and obstetricians.

Reviewer 3 Report

The author answered all the questions and improved the manuscript. Now it can be accepted for publication.

Author Response

Thank you very much for accepting the manuscript for publication after the corrections were made.